# Estimation of Intake of Critical Nutrients Associated with Noncommunicable Diseases According to the PAHO/WHO Criteria in the Diet of School-Age Children in Montevideo, Uruguay

**DOI:** 10.3390/nu14030528

**Published:** 2022-01-26

**Authors:** Florencia Köncke, Cecilia Toledo, Christian Berón, Iael Klaczko, Alicia Carriquiry, Gustavo Cediel, Fabio S. Gomes

**Affiliations:** 1Independent Researcher, Montevideo 11600, Uruguay; cecitole@gmail.com (C.T.); cberon@gmail.com (C.B.); iaelrkc@gmail.com (I.K.); 2Department of Statistics, Iowa State University, Ames, IA 50011, USA; alicia@iastate.edu; 3School of Nutrition and Dietetics, Food and Nutrition, University of Antioquia, Calle 70 No. 52-21, Medellin 050010, Colombia; gustavo.cedielg@udea.edu.co; 4Pan American Health Organization, World Health Organization, 525 23rd St. NW, Washington, DC 20037, USA; gomesfabio@paho.org

**Keywords:** NOVA, nutrient profiling, food processing, ultra-processed food, school-age food intake

## Abstract

Purpose: To estimate the effect of the consumption of products with an excessive amount of critical nutrients associated with NCDs, according to the PAHO Nutrient Profile Model on the quality of the diet of Uruguayan school-age children (4 to 12 years). Methods: A 24 h recall of food intake was conducted in a representative sample of 332 participants in the evaluation of the School Feeding Program in 2018 in public schools in Montevideo, Uruguay. Food and preparations were categorized according to the NOVA food classification, according to the nature, extent, and purposes of the industrial processes they undergo. Later, they were analyzed according to the Pan American Health Organization Nutrient Profile Model (PAHO NPM) to identify processed and ultra-processed products with an excessive content of critical nutrients. Results: Only 0.52% of children consumed exclusively natural foods, or culinary ingredients. Twenty-five per cent of children consumed ≥4 products categorized with an excessive content of free sugars, total fat, or saturated fat according to the PAHO NPM; in the case of excessive sodium, this was 40%. In general, children who included products with excessive free sugars, sodium, or saturated fat in their diet exceeded the limits established by the World Health Organization, and, as a result, their diet is of poorer nutritional quality compared to children who did not consume such products. Conclusion: Diets free of ultra-processed and processed products with excess free sugars, total fats, saturated fats, and sodium increased the chances of school-age children in Montevideo of meeting WHO nutrient intake recommendations. Meanwhile, intake of each additional gram of products with excessive critical nutrients according to PAHO NPM, significantly worsens diets, preventing children from meeting WHO recommendations.

## 1. Introduction 

The existing evidence reaffirms the need for children to eat adequately, because it is their right, and because meeting dietary guidelines results in healthier children who reach their maximum potential, and improves education outcomes [1,2,3,4,5]. The latest information available shows that the main nutritional problem of school-age children is overweight and obesity, which profoundly affects their current and future health and wellbeing [1,5,6,7,8,9]. Childhood overweight and obesity result from the imbalance between caloric intake and caloric expenditure, a situation that has been largely related to the so-called obesogenic environment. This is also part of profound changes in dietary patterns that displaced a diet based on healthy and minimally processed foods for ultra-processed products (UPP) [6,7,10,11,12]. The latter have a very low nutritional value, and their consumption has been proposed as an indicator of the nutritional quality of the diet, in particular for the study of nutrients related to noncommunicable diseases (NCDs) [12,13,14,15,16,17,18,19].

UPP contain excessive amounts of energy, fat, and sugar, and low amounts of vitamins, minerals, and other essential micronutrients for young children [5,20,21,22,23]. The definition of UPP was taken from to the “NOVA food classification”, which has been described elsewhere [22] This classification defines UPP as those products that include a large number of ingredients, many of them exclusively for industrial use, and that are the result of a sequential process in the industry [22,24]. The sale of these products in Uruguay grew by 146% between 2000 and 2013, whereas the region of the America reached a 68% increase on average. This increase in sales combined with household consumption may lead to an excessive intake of free sugars, sodium, total fat, saturated fat, and trans fatty acids [9,25].

Research carried out in multiple countries has demonstrated the relationship between UPP consumption and nutrient intake imbalances, characterized by a high concentration of nutrients associated with NCDs, and a low concentration of health-protective nutrients [10,11,12,13,14,15,16,19,24,25]. In addition, high intake of UPP has been associated with a higher incidence of hypertension, dyslipidemia, diabetes, and cancer [26,27]. Unhealthy diet is the main risk factor for NCDs [27]. In 2018, the four main NCDs were estimated to be the cause of 54.5% of all deaths in the country [28].

Previous reports of the evaluation of school feeding programs show that the average group intake exceeds the daily calorie recommendation by 13%, and 54% of children who participated have an excessive caloric intake, with one out of every three calories consumed from UPP. The average daily intake of free sugars and sodium was close to 100 g and 2000 mg, respectively. Additionally, children in the study have an insufficient intake of nutrients recommended to reach a healthy diet [1,29]. School-age children showed an average consumption of fruits and vegetables that did not reach 50% of the recommendation, which is reflected in the low intake of dietary fiber [29].

There is not enough information in Uruguay on how the intake of UPP impacts the nutrient profile in school-age children. In addition, there is insufficient information on the consumption of nutrients linked to NCDs from an early age.

The aim of the study was to estimate the effect of the consumption of products with an excessive amount of critical nutrients associated with NCDs, according to the PAHO Nutrient Profile Model (PAHO NPM) on the quality of the diet of children 4 to 12 years in Montevideo, Uruguay. Specifically, it describes: (1) the intake of products with excessive amounts of critical nutrients associated with non-communicable diseases according to the PAHO NPM; (2) the difference in the intake of critical nutrients associated with NCDs when any quantity of products with excessive critical nutrients according to the PAHO NPM compared to when no such product is consumed; (3) the difference in the prevalence of non-recommended intake levels of critical nutrients associated with NCDs, between the condition of the consumption of products with excessive content in these nutrients compared to the no-consumption condition; and (4) the contribution of the amount of products with an excessive content of critical nutrients associated with NCDs according to the PAHO MPN consumed on the total intake in the diet of these critical nutrients, with the percentage increase in this intake above the limits recommended by the WHO [20,30,31].

## 2. Materials and Methods

### 2.1. Data Sources

The data source for this analysis is the study of the Estimation of Food Intake by Multiple Pass 24-h Dietary Recall (24 h) conducted within the Evaluation of the School Feeding Program, which monitored the nutritional status of children of public and private elementary schools in Uruguay in 2018. The research project was conducted by a specially trained nutritionist team. During the fieldwork, a licensed nutritionist was in the school canteen when the children ate lunch, and observed and quantified the food intake [1,32]. 

The estimation of food intake was carried out in a representative sample of 21 public elementary schools in the city of Montevideo, which had a school canteen service. The children evaluated correspond to students of initial level (pre-school children), second and fifth grade, selected by multistage sampling in the second semester of the year. In order to estimate intake, we included what was consumed in the 24-h period by the selected children, that is, what was consumed at school and at home. The entire team was trained in the application of this technique, and in the use of the available materials. The survey was carried out by licensed nutritionists or advanced nutrition students, trained on the application of the tool, and in the use of the available materials to quantify the intake. Three-hundred and thirty-two first 24 h were performed, and a second was applied to 18% of the sample [32,33]. The methodology of the study is described in the document of the Division of Research, Evaluation and Statistics (DIEE) [1]. This study was based on the analysis of the first 332 24 h.

### 2.2. Classification of Foods According to Their Processing

The NOVA classification system groups food according to the nature, extent, and purposes of the industrial processes they undergo. This system classifies foods and food products into four groups: group 1—unprocessed and minimally processed foods (G1); group 2—processed culinary ingredients (G2); group 3—processed foods (G3); group 4—ultra-processed foods (G4) [22,24].

According to the PAHO NPM, for this study, the NOVA classification was adapted only with groups NOVA 3 (G3) and NOVA 4 (G4), according to the following:

Group 3: includes canned, bottled, or brined foods and culinary preparations that are prepared following traditional methods, using unprocessed or minimally processed foods, and are made by adding excessive culinary ingredients, such as salt, sugar, butter, or oil, or even preservatives. Some examples of foods in G3 are cheese, fruits, and vegetable preserves. 

Group 4: ultra-processed foods, which includes foods, beverages, and other industrial preparations designed and marketed by the food industry. This implies adding artificial additives from G3 or formulations and ingredients used in processed foods, such as sugar, oils, fats, or salt, and other sources of energy and nutrients that are not normally used in culinary preparations, or other products of additional processing, such as hydrogenation, interesterification of oils, protein hydrolyzation, or food components, such as soy protein isolate, maltodextrins, inverted sugar, sugar syrup, and other additives that mimic or increase the sensory characteristics of foods, or that decrease less desired characteristics in the final product. Production of UPP requires a variety of processes in sequence, that do not have domestic equivalence, to combine the several ingredients and get to the final product. Some examples of products in G4 are sugar-sweetened beverages, candy, ice cream, and sweet and savory snacks. In this study, we use ultra-processed foods and ultra-processed products as synonyms [12,22].

#### 2.2.1. Criteria Utilized for Harmonization of Records Obtained by 24 h with the NOVA Classification

A criterion was created to differentiate homemade or restaurant preparations based on unprocessed or minimally processed foods from those that have similar characteristics, but are industrially made. 

Homemade culinary or traditional preparations are made from foods in G1, but similar processed products of such preparations with added salt, sugar, butter, or oil, as well as additives and antioxidants were classified as G3. If those products were added with artificial additives to mimic the sensory qualities of G1, or if the product contains high quantities of ingredients in G4, or are products designed or marketed industrially, they were classified as G4.

Foods and beverages that must be evaluated with the PAHO nutrient profile model are processed and ultra-processed products, and there is no evidence to support the need to apply it to unprocessed or minimally processed foods. According to the PAHO NPM, processed and ultra-processed products are classified as follows: with an excessive amount of sodium, if the ratio between the amount of sodium (mg) in any given amount of the product and the energy (kcal) is equal to or greater than 1:1; with an excessive amount of free sugars, if in any given quantity of the product the amount of energy (kcal) from free sugars (grams of free sugars × 4 kcal) is equal to or greater than 10% of the total energy (kcal); with an excessive amount of total fat, if in any given quantity of the product the amount of energy (kcal) from total fat (grams of total fat × 9 kcal) is equal to or greater than 30% of total energy (kcal); with an excessive amount of saturated fat, if in any given quantity of the product the amount of energy (kcal) from saturated fat (grams of saturated fat × 9 kcal) is equal to or greater than 10% of the total energy (kcal) [34].

#### 2.2.2. Assessment of Energy and Nutrient Input of Foods 

Uruguay does not have an updated and accessible food composition table. In this scenario, an ad hoc table was generated, based on a pre-existing composition table to which food, preparations, and products commonly consumed by school-age children were added. 

For all foods, the old Uruguayan Food Composition Table was used as the first source of information. When it lacked data, the Brazilian Table of Food Composition was used and, thirdly, the Spanish Table of Food Composition [1]. 

For preparations commonly consumed by children, the School of Nutrition (EN) of the University of the Republic (UdelaR) analyzed the chemical composition of standardized preparations per 100 g, and added the information from the recipes of the PAE.

In the case of industrialized foods and products, the information provided by the companies on the web or the nutritional composition on the packaging was used. 

In this study, only the dietary energy intake and the contribution of nutrients related to NCDs, total and saturated fat, free sugars, and sodium are analyzed.

#### 2.2.3. Data Processing and Analysis 

To estimate nutrient intake using the 24 h data, food was transformed into their respective nutrients using the Diet Intake Evaluation Program (EVINDI v4) of the School of Nutrition and Dietetics of the University of Antioquia. For this, and previously, the database of preparations, foods, and beverages consumed in Uruguay, and representative data of the chemical composition were updated. The database obtained was transformed to Stata/SE 12.0 where it was linked to sociodemographic variables (sex, age group, and educational level of the adult informant), defined as covariates for subsequent analysis. 

Analysis includes descriptive statistics, including the prevalence of intake of products with an excess of critical nutrients, averages of the proportion of energy intake from each critical nutrient and sodium intake in mg, and the prevalence of critical nutrient intake according to the WHO intake goals, as well as the respective 95% confidence intervals. The averages of intake of critical nutrients (expressed as a proportion of energy, or as a ratio of mg of sodium/kcal) and the prevalence of intake of nutrients above or below the WHO intake goals were estimated for the total population, and for the domains of interest, that is, for the subset of the population that did not consume, and that consumed excessive products according to PAHO. 

Linear regression models were adjusted to estimate the significance of the contribution of the consumption of products with an excess in free sugars, total fats, and saturated fats above the average intake of the proportion of energy in the diet from each of these critical nutrients, adjusting by sociodemographic variables (i.e., age, sex, and educational level). To estimate the significance of the contribution of the consumption of products with excessive sodium, sodium intake measured in mg was included in the linear regression models as a response variable. The statistics considered the structural information of the sample design.

Prevalence ratios were estimated to compare the proportions of the population and subsets of interest that did not meet the WHO nutrient intake goals. Logistic regression models with a Poisson probit link function, adjusted for the sociodemographic variables, were used to estimate the significance of the contribution of the consumption of products with an excess in critical nutrients according to the PAHO NPM, and the probability of the intake of the critical nutrients being above the goals recommended by the WHO. Finally, linear regression models were adjusted to analyze the significance of the contribution of grams of products with excessive nutrients on the distance from the intake of critical nutrients to that of the WHO goals.

## 3. Results

The sample consisted of 332 children between 4 and 12 years of age, from 21 public schools in the city of Montevideo with canteen services of the school feeding program. The children and the days on which the 24-h reminders were collected were randomly selected. Of the total number of children, 88 attended preschool (47 males and 41 females), 114 attended second grade (64 males and 50 females), and 130 attended fifth grade (65 males and 65 females). The mean age was: 4.5 years in preschool, 7.9 years in second grade, and 10.8 years in fifth grade. Overweight and obesity was observed more in boys than in girls, 36.4% vs. 35.1% respectively.

Table 1 shows the consumption of foods categorized for which the PAHO NPM does not apply, that is, the group of children who exclusively consumed natural, unprocessed, and minimally processed foods or culinary ingredients; the group that consumed ultra-processed or processed products without an excessive content of one of the nutrients related to NCDs according to the PHAO NPM; and those that consumed products that were excessive in the nutrients studied. The proportion of children who did not include products defined as not excessive in any of the nutrients evaluated was 5% (Table 1).

On the other hand, more than half of girls and boys consumed products with an excessive content of nutrients associated with NCDs. It is noteworthy that a fifth of the school-age children included at least one product with excessive free sugars, and the same proportion consumed products with excessive total fat. When analyzing the prevalence of the consumption of multiple products with excessive amounts of critical nutrients, it was identified that 61.3% of children between 4 and 12 years consumed three or more products that were excessive in sodium. In the case of free sugars or total fats, this proportion reached about 50% (Table 1).

From the analysis of the caloric intake provided by the studied nutrients to the total energy intake for the entire population studied, it is observed that 18.2% of the total energy comes from free sugars. In the population that consumes products with excessive content according to the PAHO NPM, it was observed that 18.8% of total energy comes from free sugars. The contribution of free sugars to the dietary energy is 4.9% in the group of children who did not consume products with an excessive content of critical nutrients (Table 2).

The contribution of total fat to the total energy of the diet for the whole population studied is within the goal of intake recommended by the WHO, and is similar among the subgroup of the population that consumed products with excessive total fat. Despite this, the adequacy improves significantly when there is no consumption of food products with excessive total fat according to the PAHO NPM. The proportion of energy from this nutrient is 4.4 percentage points lower (*p* < 0.01) among those who did not consume products with excessive fat. (Table 2).

Saturated fat intake exceeds that recommended by the WHO in school-age children who consumed ultra-processed or processed products with an excess of this critical nutrient (Table 2), being significantly lower (*p* < 0.01), and meeting recommendations among those children that included only foods without excessive content in this nutrient according to PAHO NPM. 

The average sodium intake of school-age children studied exceeds the limits established by the WHO for the intake of this nutrient. The population group that included products with excessive sodium exceeded the established limit, and the population that did not include such products in their diet had a significantly lower (*p* < 0.01) and adequate sodium intake (Table 2).

The prevalence of inadequate nutrient intake levels associated with NCDs indicates that 75.4% of children exceed the WHO recommended limit of 10% or less of total daily dietary energy from free sugars; 39.9% do not meet the recommendation for total fat; 57% do not meet the recommendation for saturated fat; and 56.6% do not meet the sodium recommendation (Table 3).

When this analysis is carried out by fractions of the population according to the presence or not of products with excessive amounts of these nutrients in the diet, a higher prevalence of excessive intake is observed among those who consume products with excess in these nutrients according to PAHO.

The highest prevalence of inadequate intake is found among school-age children who consume products with an excess in free sugars. About four out of five school-age children who consume products with excessive free sugars do not meet the WHO recommendation. On the other hand, the prevalence of school-age children with an inadequate intake of free sugars, saturated fats, and sodium is significantly lower among those who have a diet free of products with excessive amounts of these nutrients according to the PAHO MPN (Table 3).

When analyzing the association between the consumption of products with an excessive content in grams of fat, saturated fat, free sugars, and milligrams of sodium with the content of critical nutrients in the diet, and with the prevalence of the non-recommended intake of these nutrients, a dose effect response was observed, in which the probability of inadequate consumption of these nutrients, and excess intake above the limit established by the WHO, are associated with an increase in the consumption of products with excessive content.

As seen in Table 4, the consumption of each additional gram of products that contain excessive amounts of free sugars, total fat, saturated fat, or sodium according to the PAHO NPM results in not meeting WHO intake goals. For each gram of product with excessive free sugars according to PAHO, intake moves away from the WHO intake goal by 0.0002 percentage points. Additionally, each gram of product with excessive total fat, and each gram of product with excessive saturated fat, moves away from the WHO intake goal by 0.0002 and 0.00001 percentage points, respectively.

Each gram of product with excessive sodium according to PAHO increases dietary sodium intake by 4.67 mg above that recommended by the WHO. This means that the intake of any amount of products with excessive critical nutrients, according to PAHO, generated a significant negative impact on the diet, moving it away from what is recommended by the WHO.

## 4. Discussion

The results of this study show that Montevideo school-age children who consume ultra-processed and processed products with excessive amounts of free sugars, total fats, saturated fats, or sodium, according to the PAHO-NPM, have a diet with poorer nutritional quality, and a greater probability of not meeting the WHO recommendations for critical nutrients related to NCDs compared to school-age children who do not consume such products.

Currently, school-age children’s diet exceeds the national caloric goals for this age group [1]. In addition, this study confirms previous findings that indicate an imbalance in relation to the intake of nutrients that are associated with NCDs, the main cause of death and disability in the country. We found that the energy intake from free sugars is almost two times the WHO recommended threshold [20,35].

The PAHO nutrient profile model has been developed as a food classification tool to develop public policies to reduce the demand and supply of ultra-processed and processed food products in order to help populations achieve critical nutrient intake goals established by the WHO. Multiple studies with different populations in various countries of the world have confirmed that a higher consumption of ultra-processed products is associated with a poorer diet quality, exceeding limits of nutrients associated with NCDs, such as free sugars, total fats, saturated fats, and sodium, and contributing less than the recommended intake for fiber, vitamins, and essential minerals [15,16,21,23,24,25,36,37,38,39,40,41].

Likewise, the consumption of processed products with excessive amounts of fat, saturated fat, trans fat, sugar, and sodium are also associated with noncommunicable diseases and all-cause mortality [42,43,44,45]. Our study shows that the consumption of products with excessive critical nutrients explains the imbalance in the diet of school-age children in Montevideo in relation to the WHO dietary recommendations. Half of the children who consumed ultra-processed and processed products with an excess of some of the nutrients linked to the development of NCDs included three or more of these products in their diet. Consequently, from the study of dietary intake, it was found that those children who consumed ultra-processed and processed products with an excessive content of these nutrients according to the PAHO NPM present greater dietary inadequacy due to excess in the consumption of these nutrients.

Excess weight, and the consumption of ultra-processed and processed products that impact dietary intake of the population, present a complex scenario to overcome. At the international political level, there is consensus on the urgent need to implement population-based measures that aim at reducing the impact that unhealthy diet has on childhood [20]. Undoubtedly, the great challenge is to improve the diet of children, promoting dietary practices that are based on fresh or minimally processed foods, and reducing the consumption of processed and ultra-processed food products with excessive sugars, fats, and sodium, which will result in better growth and development, and allow children to achieve their full potential. Uruguay faces a great challenge in terms of public policies related to health protection and healthy diets, especially with regard to children.

International agencies, such as PAHO/WHO, UNICEF, and FAO, have repeatedly pointed out the need to generate multisectoral strategies to reduce the health and economic impact that this situation will have on the entire population, but especially on children of all ages [46]. Furthermore, following international recommendations, Uruguay has made recent progress in the development of policies, actions, and strategies for national application. On the one hand, the Dietary Guidelines for the Uruguayan population, published by the Ministry of Public Health [12], contains recommendations and messages that seek to promote the application of dietary practices by the general population that are compatible with achieving healthy and sustainable diets and food systems. The Dietary Guidelines promote a diet based on natural foods and avoiding the consumption of ultra-processed products. On the other hand, progress has been made in inter-institutional consensuses that seek to protect the health of children through the promotion of healthy dietary practices, integrated transversally into the life and educational trajectory of all children [3,8,47]. Additionally, in February 2021, Uruguay began the application of front-of-package nutritional warning labeling on food products that included modifications to the nutrient profile model established in the original regulations approved in 2018 [48]. The regulations in force in Uruguay established looser limits that exempt more products with an excess of critical nutrients than what is recommended by PAHO from the obligation to apply warnings on the front of the package [2,49].

Considering the progress in Uruguay, and the results of this study, there are still adjustments to be made to the policies adopted by the country to guarantee that all the products with excessive nutrients associated with NCDs according to the PAHO NPM are regulated, and that the regulatory measures on labeling, school environments, and others are aligned with WHO recommendations

The dose–response effect found in this study reveals that the intake of any amount of products classified as excessive in nutrients of public health concern, according to the PAHO NPM, compromise reaching a healthy diet. The sale of ultra-processed products is on the rise in Latin America [9,25], and between 2000 and 2013, Uruguay presented the highest increase in sales of ultra-processed products in the region, with 7.2% growth per year [25]. Compared to 2000, 243 g more of ultra-processed products were sold per capita per day in 2013. Taking this figure, it could be estimated, based on the result of the dose–response effect of this study, that each additional contribution to the diet of 243 g of ultra-processed products that are excessive in sugars, fats, saturated fats, and sodium would make it 44.7% more excessive in free sugars, 14.9% more excessive in fats, 18.5% more excessive in saturated fats, and 56.7% more excessive in sodium, according to WHO recommendations.

The main limitations of the study are: (a) Uruguay does not have an updated food composition table, and due to this, the intake was estimated considering a compilation of multiple sources; (b) there is insufficient information to estimate the intake of trans fat; (c) the representativeness of the sample; (d) intake of foods and beverages analyzed here is representative of the spring season (survey in October and November).

The main strengths of the study can be summarized as: (a) the data collection was carried out by licensed nutritionists or advanced students trained specifically for this study; (b) part of the information was collected by direct observation, minimizing the bias associated with the recall of the interviewee; (c) the food composition table includes an estimation of nutrients in foods and preparations of usual consumption carried out by the School of Nutrition of the UdelaR, as well as all the preparations provided by the school feeding program.

In conclusion, this study shows that policies to reduce demand and supply of products which aim at improving the diet of populations, including school-age children, and protecting public health, will be better aligned with those recommended by the WHO if they adopt the PAHO NPM. Diets free of ultra-processed and processed products with excess free sugars, total fats, saturated fats, and sodium were the best option for school-age children in Montevideo, increasing their chances of meeting WHO recommendations. Meanwhile, intake of products with excessive critical nutrients according to PAHO (and each additional gram consumed of such products) significantly worsens diets, preventing children from meeting WHO recommendations.

## Figures and Tables

**Table 1 nutrients-14-00528-t001:** Consumption of products defined as excessive in NCD-related critical nutrients according to PAHO-NPM ^‡^.

% of Individuals with Consumption of:	Free Sugars	Total Fat	Saturated Fats	Sodium
	% (95% CI)	% (95% CI)	% (95% CI)	% (95% CI)
Only foods that do not apply to the PAHO NPM (a)	0.52	0.52	0.52	0.52
	(−0.5–1.5)	(−0.5–1.5)	(−0.5–1.5)	(−0.5–1.5)
Products defined as non-excessive in NCD-related critical nutrients according to PAHO NPM (b)	4.4	8.6	10.4	3.7
	(1.5–7.2)	(5.1–12.2)	(6.4–14.2)	(1.3–6.1)
Products defined as excessive in NCD-related critical nutrients according to PAHO NPM (c)				
1 product	21.2	22.0	19.5	12.3
	(16.3–27.0)	(16.9–29.0)	(14. 8–25.3)	(8.7–17.0)
2 products	25.8	22.6	21.3	22.2
	(20.5–32.0)	(17.7–28.5)	(16.4–27.0)	(17.2–28.1)
3 products	22.3	19.9	23.5	21.7
	(17.4–28.1)	(15.2–25.5)	(18.4–29.6)	(16.9–27.5)
≥4 products	25.7	26.4	24.8	39.6
	(20.8–31.4)	(21.2–32.2)	(19.9–30.5)	(33.5–46.0)

^‡^ Assessment of dietary intake. School feeding program children 4/12 years. Total population estimate: *N* = 15,070 (*n* = 332). 95% CI: 95% confidence interval; NCD: noncommunicable diseases; PAHO-NPM: Pan American Health Organization Nutrient Profile Model. (a) Foods that do not apply to the PAHO NPM: natural of minimally processed foods and culinary processed ingredients. (b) Foods that apply to the PAHO NPM without a high content of NCD-related nutrients according to the PAHO-NPM (for free sugars < 10% of total energy, for total fat < 30% of total energy, for saturated fats < 10% of total energy, for sodium < 1 mg per kcal). (c) Processed and ultra-processed products with excessive content of NCD-related nutrients according to PAHO-NPM (for free sugars ≥ 10% of total energy, for total fat ≥ 30% of total energy, for saturated fats ≥ 10% of total energy, for sodium ≥ 1 mg per kcal).

**Table 2 nutrients-14-00528-t002:** Mean intakes of NCD-related critical nutrients in the overall population, and in fractions of the population consuming products with and without an excessive content in NCD-related critical nutrients according to the PAHO NPM ^‡^.

Nutrient Dietary Content	Overall Diet		With Excessive Content of NCD-Related Critical Nutrients ^a,b^		Without Excessive Content of NCD-Related Critical Nutrients ^c^		Coef. (95% CI) ^d^	
Nutrient Dietary Content	Mean (95%)	(95% Conf. Interval)	Mean (95%)	(95% Conf. Interval)	Mean (95%)	(95% Conf. Interval)		(95% Conf. Interval)
Free sugars (% of total energy intake)	18.2	(16.8–19.6)	18.8	(17.4–20.2)	4.9	(3.2–6.7)	13.8 *	(11.6–16.0)
Total fats (% of total energy intake)	29.4	(28.5–30.3)	29.7	(28.9–30.6)	26.0	(21.8–30.2)	4.4 **	(2.7–8.5)
Saturated fats (% of total energy intake)	11.0	(10.5–11.4)	11.2	(10.7–11.6)	9.5	(7.7–11.4)	1.8 **	(1.1–3.6)
Sodium (mg) Children between 5 and 10 years	1910.6	(1773.7–2047.6)	1939.8	(1799.3–2080.2)	1242.6	(933.1–1552.0)	743.7 *	(449.3–1038.1)

^‡^ Assessment of dietary intake. School feeding program, children 4/12 years. Total population estimate: *N* = 15,070 (*n* = 332). Children 4, 11, and 12 years old were excluded for this analysis. 95% CI: 95% confidence interval; NCD: noncommunicable diseases; PAHO-NPM: Pan American Health Organization Nutrient Profile Model. ^a^ Linear regression for sodium. ^b^ Products with excessive content of NCDs related nutrients according to PAHO-NPM (for free sugars ≥ 10% of total energy, for total fat ≥ 30% of total energy, for saturated fats ≥ 10% of total energy, for sodium ≥ 1 mg per kcal). ^c^ Foods that do not apply to the PAHO NPM (unprocessed, minimally processed foods and culinary ingredients) and ultra-processed and processed products without high content of NCDs related nutrients according to PAHO-NPM (for free sugars < 10% of total energy, for total fat < 30% of total energy, for saturated fats < 10% of total energy, for sodium < 1 mg per kcal). ^d^ Lineal regression models adjusted age groups, sex, education. Significant difference between the fraction of the population with diets made up of products with versus without excessive content of NCD-related critical nutrients according to PAHO NPM: * *p* ≤ 0.001, ** *p* ≤ 0.05.

**Table 3 nutrients-14-00528-t003:** Assessment prevalence of non-recommended intake levels of NCD-related critical nutrients ^†^ in the whole population, and by fractions of population diet with and without products with excessive content in these critical nutrients ^‡^.

Fraction of the Population Diet Made Up of Products												
NCD-Related Critical Nutrient	Whole Population		With Excessive Content of NCD-Related Critical Nutrients ^a,b^		Without Excessive Content of NCD-Related Critical Nutrients ^c^		Coef. (95%) ^d^		PR (95% CI) ^e^			
	Individuals Who Did not Meet the Recommendation (%) ^†^						Adjusted ^§^	(Conf. Interval)	Crude	(Conf. Interval)	Adjusted ^§^	(Conf. Interval)
	Mean (95%)	(Conf. Interval)	Mean (95%)	(Conf. Interval)	Mean (95%)	(Conf. Interval)						
Free sugars	75.7	(69.2–80.6)	78.0	(71.9–83.1)	24.3	(6.2–60.8)	1.48 *	(0.6–2.3)	3.21 *	(1.1–9.3)	3.2 **	(1.1–9.3)
Total fat	39.9	(33.8–46.3)	40.4	(34.0–47.1)	35.0	(16.9–58.8)	1.99	(−0.9–3.2)	1.15	(0.6–2.1)	1.2	(0.6–2.2)
Saturated fats	57.4	(51.0–63.6)	60.1	(53.3–66.6)	35.0	(18.3–56.4)	0.67 **	(0.1–1.2)	1.72	(0.98–2.98)	1.7 **	(1.0–2.99)
Sodium	56.7	(50.3–62.8)	58.7	(52.2–49.0)	10.5	(9.9–58.2)	1.49 **	(0.4–2.6)	5.57	(0.9–35.7)	5.5	(0.9–34.4)

^‡^ Assessment of dietary intake. School feeding program (PAE), children 4–12 years. Total population estimate: *N* = 15,070 (*n* = 332). 95% CI: 95% confidence interval; NCD: non communicable diseases; PAHO-NPM: Pan American Health Organization Nutrient Profile Model. ^†^ WHO Recommended intake of NCD-related critical nutrients: for free sugars <10% of total energy, for total fat < 30% of total energy, for saturated fats < 10% of total energy, for sodium (mg) content recommended value per age group: <2000 mg (for adults and adolescents); <1640 mg (for children aged 5–10 years old); and <1122 mg (for children aged <5 years old)). Sodium intake recommendation for children were derived from average energy requirements for children with moderate physical activity level, as estimated by FAO/WHO/UNU). Guideline: sodium intake for adults and children. World Health Organization; 2012; Human Energy Requirements. FAO/WHO/UNU; 2004. ^a^ Values are percentages derived from Probit regression models. Significant difference between the fractions of the population with diets made up of products with versus without excessive content of NCD-related critical nutrients, * *p* ≤ 0.001, ** *p* ≤ 0.05. ^b^ Products with excessive content of NCD-related critical nutrients according to PAHO NPM (for free sugars ≥10% of total energy, for total fat ≥ 30% of total energy, for saturated fats ≥ 10% of total energy, for sodium ≥ 1 mg per kcal). ^c^ Foods that do not apply to the PAHO NPM products, and without a high content of NCD-related nutrients according to the PAHO nutrient profile (for free sugars < 10% of total energy, for total fat < 30% of total energy, for saturated fats < 10% of total energy, for sodium < 1 mg per kcal). ^d^ Probit regression models. ^e^ PR, prevalence ratios estimated using Poisson regression. ^§^ Adjusted age, sex, education level, and family income.

**Table 4 nutrients-14-00528-t004:** Association between consumption of products with excessive content of NCD-related critical nutrients (grams) according to PAHO NPM, and the content provided by the respective nutrients ^†^, and the prevalence of non-recommended intake of these nutrients in the whole diet ^‡^.

	Effect Size over the Content of NCD-Related Critical Nutrients ^†^		Prevalence of Non-Recommended Intake of NCD-Related Critical Nutrients ^††^	
	Coef. (95%) ^a^	(Conf. Interval)	PR (95%) ^b^	(Conf. Interval)
Free sugars	0.0001841 *	0.0001357	1.000005 *	1.000002
		0.0002326		1.000007
Total fat	0.0001839 *	0.0001207	1.000018 *	1.000011
		0.0002472		1.000025
Saturated fats	0.000076 *	0.0000465	1.000013 *	1.000008
		0. 0001054		1.000018
Sodium	4.670705 *	3.5052	1.000018 *	1.000012
		5.83621		1.000024

^‡^ Assessment of dietary intake. School feeding program (PAE), children 4/12 years. Total population estimate: *N* = 15,070 (*n* = 332). 95% CI: 95% confidence interval; NCD: noncommunicable diseases; PAHO-NPM: Pan American Health Organization Nutrient Profile Model. ^†^ Content of NCD-related critical nutrients: free sugar, total fat, saturated fat (% of the total energy intake). Sodium content: total sodium (mg) less than the recommended value per age group according to WHO (2000 mg (for adults and adolescents); 1640 mg (for children aged from 5 to 9.9 years old); and 1122 mg (for children aged <5 years old)). Guideline: sodium intake for adults and children, World Health Organization, 2012. ^††^ For free sugar ≥ 10% of total energy intake, for total fat ≥ 30% of total energy intake, for saturated fat ≥ 10% of total energy intake, for sodium ≥ 2000 mg (for adults and adolescents); ≥1640 mg (for children aged from 5 to 9.9 years old); ≥1122 mg (for children aged <5 years old). Sodium intake recommendation for children were derived from average energy requirements for children with moderate physical activity level, as estimated by FAO/WHO/UNU) Guideline: sodium intake for adults and children, World Health Organization, 2012. ^a^ Lineal regression models for free sugar, total fat, saturated fat intake, and linear regression models for sodium adjusted for age groups, sex, education level, and family income. * *p* ≤ 0.001. ^b^ PR, prevalence ratios estimated using Poisson regression models adjusted for age, sex, education level, and family income. * *p* ≤ 0.001.

## Data Availability

https://evaluacionpae.anep.edu.uy/ (accessed on 4 January 2022).

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
