# Peer review of "Estimation of Intake of Critical Nutrients Associated with Noncommunicable Diseases According to the PAHO/WHO Criteria in the Diet of School-Age Children in Montevideo, Uruguay"

_nutrients, 2022, doi:10.3390/nu14030528_

Round 1

Reviewer 1 Report

This is an interesting study that attempts to evaluate the dietary intake of school-age children Montevideo, Uruguay. There is some improvement required with grammar and structuring of sentences but the research design is adequate. The Introduction needs improvement especially in the final two paragraphs so that the study is prospective and not being introduced retrospectively. There is inconsistency between the past and present tense when describing the results plus at other places throughout the manuscript. Further comments are provided below.

Abstract

Not “problem in the region of the Americans” but “problem in America”

Remove any references from this section.

Define “NOVA”

Introduction

Not “problem in the region of the Americans” but “problem in America”

Define “NOVA”

“higher incidence of hypertension, dyslipidaemia…”

“The average group intake in children that participated in this study..” – what study are you referring to?

The following sentence is incomplete” “insufficient consumption of foods recommended by dietary.”

“critical nutrients” – should be “critical nutrients”

Define “PAHO NPM” and “PAHO MPN”

Do not include any results of the present study in the Introduction! The information cover you be prospective.

Materials and Methods

“24-HOUR dietary recall”

“level 4 years” – what does that mean? How old were the students?

“According to the parameters…”

Please correct the following: “using unprocessed of minimally processed”

“formulation of Ingredients”

“similar characteristics”

Please revise the following: “If to those products were added…”

Results

Please be consistent and describe results in the past-tense.

Table 2 – “…defined according to the presence or absence of products..”

“Fractions of the population according to the presence”

“Critical nutrients in the diet”

“Dose-response effect”

“PAHO increases dietary sodium”

Discussion

“have have repeatedly” – delete a “have”

“systemsguide” – please correct

What does the following mean “UdelaR”?

Author Response

Abstract

Not “problem in the region of the Americans” but “problem in America”,

corrected

Remove any references from this section.

done

Define “NOVA”,

done. “It groups all foods according to the nature, extent and purposes of the industrial processes they undergo”

 Introduction

Not “problem in the region of the Americans” but “problem in America”,

corrected

Define “NOVA”,

done. “It groups all foods according to the nature, extent and purposes of the industrial processes they undergo”

 “higher incidence of hypertension, dyslipidaemia…”

done

“The average group intake in children that participated in this study..” – what study are you referring to?

Done. “The aim of the study was…”

The following sentence is incomplete” “insufficient consumption of foods recommended by dietary.”

Done. Additionally, children in the study have an insufficient intake  of nutrients  recommended to reach a healthy diet 

“critical nutrients” – should be “critical nutrients”,

corrected

Define “PAHO NPM” and “PAHO MPN”,  

done “PAHO MPN”

Do not include any results of the present study in the Introduction! The information cover you be prospective.

Ok

Materials and Methods

“24-HOUR dietary recall”

corrected

“level 4 years” – what does that mean? How old were the students?

corrected

“According to the parameters…”

corrected

Please correct the following: “using unprocessed of minimally processed”

corrected

“formulation of Ingredients”

corrected

“similar characteristics”

corrected

Please revise the following: “If to those products were added…”

corrected

Results

Please be consistent and describe results in the past-tense.  

corrected

Table 2 – “…defined according to the presence or absence of products..”

Done, Table 2. Mean intakes of NCD-related critical nutrients in the overall population and in fractions of the population consuming products with and without excessive content in NCD-related critical nutrients according to the PAHO NPM

“Fractions of the population according to the presence” corrected

“Critical nutrients in the diet” corrected

“Dose-response effect” corrected

“PAHO increases dietary sodium” corrected

 Discussion

“have have repeatedly” – delete a “have” corrected

“systemsguide” – please correct corrected

What does the following mean “UdelaR”? corrected, Universidad de la República.

I send in the document the changes to your comments and suggestions. 
We hope that this second version complies with your comments, 
greetings, atte

Reviewer 2 Report

The manuscript entitled “Estimation of intake of critical nutrients associated with noncommunicable diseases according to the PAHO/WHO criteria in the diet of school-age children in Montevideo, Uruguay” presents interesting issue but some problems must be corrected.

Major:

  1. Authors present a number of basic and even very trivial information that should not be presented in a scientific manuscript (e.g. “Unhealthy diet is an important health problem”) – Authors should be aware that they do not prepare the basic manual for students, or column of the newspaper, but a scientific paper that should be interesting for researchers from the area of food and nutritional sciences, so they should understand that their readers will have the nutritional knowledge.
  2. The manuscript should be prepared while using a brief and communicative sentences, without unnecessary forms and statements (e.g. “Unhealthy diet is an important health problem in the region of the Americas, and Uruguay does not escape this reality”).
  3. Typos should be corrected (e.g. “dieetary”)
  4. The manuscript is not prepared according to the instructions for authors and Authors do not present necessary information in each section – e.g. in Abstract Authors present literature references, but we do not know what are their results (it is not clearly indicated if they present here only literature data or their own results); in Patents section, Authors present instructions for authors, not their own text
  5. Authors prepared their manuscript shabbily, and readers even do not know how to read presented values – Authors present their results while using various systems – either as “0,0001”, or as “.0001”, while I believe that in all cases they meant “0.0001”
  6. It seems that Authors refer improper references – e.g. for information about School Feeding Program and monitoring of the nutritional status of children of public and private elementary schools in Uruguay in 2018, they state that “The methodology of the study is described elsewhere” and they refer the study published in 2004, conducted in a population of men (Conway J., e. a. Accuracy of Dietary Recall Using the USDA Five-Step Multiple-Pass Method in Men: An Observational Validation Study. J Am Diet Assoc. 2004;(104(4)):595-603.) – totally different population, country, and the study conducted over 10 years earlier
  7. Some references presented in the text are not included to the References Section (Köncke, Toledo, 2021)

Abstract:

Reader even do not know if Authors present the results of their own study or literature data. Authors should prepare this section properly according to the instructions for authors, without any literature references

Introduction:

Authors failed to justify the need for their study – they should present what is already known and what are the “gaps” in the scientific knowledge to formulate the aim of their study.

Instead of what was done, Authors should present the aim of the study (e.g. “The aim of the study was…”)

Materials and Methods:

Authors should present necessary details associated with applied methodology, including 24HR (how was it conducted), but also the other stages of the applied procedure

Authors should describe clearly NOVA classification

Results:

Authors should present characteristics of the studied participants and they should verify their representativeness

Authors should present the raw data, to describe clearly the studied group

In some tables Authors do not present the data which are indicated in the description – e.g. Mean [95% CI], but there is presented only mean (I suppose), without CI

Authors should not reproduce in the text data which are already presented in tables

Discussion:

Authors should: (1) compare gathered data with the results by other authors, (2) formulate implications of the results of their study and studies by other authors, (3) formulate the future areas which should be studied.

The limitations should be broadened and deepened.

Conclusions:

The conclusions from the study should be presented

Author Response

Major:

  1. Authors present a number of basic and even very trivial information that should not be presented in a scientific manuscript (e.g. “Unhealthy diet is an important health problem”) – Authors should be aware that they do not prepare the basic manual for students, or column of the newspaper, but a scientific paper that should be interesting for researchers from the area of food and nutritional sciences, so they should understand that their readers will have the nutritional knowledge.
  2. The manuscript should be prepared while using a brief and communicative sentences, without unnecessary forms and statements (e.g. “Unhealthy diet is an important health problem in the region of the Americas, and Uruguay does not escape this reality”).
  3. Typos should be corrected (e.g. “dieetary”)
  4. The manuscript is not prepared according to the instructions for authors and Authors do not present necessary information in each section – e.g. in Abstract Authors present literature references, but we do not know what are their results (it is not clearly indicated if they present here only literature data or their own results); in Patents section, Authors present instructions for authors, not their own text
  5. Authors prepared their manuscript shabbily, and readers even do not know how to read presented values – Authors present their results while using various systems – either as “0,0001”, or as “.0001”, while I believe that in all cases they meant “0.0001” Corrected, we use 0,0001
  6. It seems that Authors refer improper references – e.g. for information about School Feeding Program and monitoring of the nutritional status of children of public and private elementary schools in Uruguay in 2018, they state that “The methodology of the study is described elsewhere” and they refer the study published in 2004, conducted in a population of men (Conway J., e. a. Accuracy of Dietary Recall Using the USDA Five-Step Multiple-Pass Method in Men: An Observational Validation Study. J Am Diet Assoc. 2004;(104(4)):595-603.) – totally different population, country, and the study conducted over 10 years earlier. Corrected
  7. Some references presented in the text are not included to the References Section (Köncke, Toledo, 2021)  Corrected

Abstract:

Reader even do not know if Authors present the results of their own study or literature data. Authors should prepare this section properly according to the instructions for authors, without any literature references

Abstract:

Purpose: To explore intake levels of some specific nutrients related to non-communicable diseases in Uruguayan school-age children ( 4 to 12y), employing the PAHO Nutrient Profile Model.

Methods: Data of 332 participants that participated in the evaluation of the School Feeding Program in 2018. A 24h recall of food intake was conducted in a representative sample of public schools in the city of Montevideo. Food and preparations were categorized according to NOVA classification system. It groups all foods according to the nature, extent and purposes of the industrial processes they undergo. Later they were analyzed according to the Pan American Health Organization nutrient profile model (PAHO NPM). to identify processed and ultra-processed products with excessive content of critical nutrients

Results: Only 0.52% of children consumed exclusively natural foods, or culinary ingredients. Twenty-five per cent of children consumed ≥ 4 products categorized with excessive content of free sugars, total fat or saturated fat according to the PAHO NPM; in the case of excessive sodium, this was 40%. In general, children who included products with excessive free sugars, sodium or saturated fat in their diet exceeded the limits established by PAHO and as a result, their diet is of poorer nutritional quality compared to children who did not consume such products.

Conclusion: Diets free of ultra-processed and processed products with excess free sugars, total fats, saturated fats and sodium were the best option for school-age children in Montevideo, increasing their chances of reaching WHO recommendations. Meanwhile, intake of products with excessive critical nutrients according to PAHO NPM, and each additional gram consumed of such products, significantly worsens diets, preventing them from meeting WHO recommendations.

 Introduction:

Authors failed to justify the need for their study – they should present what is already known and what are the “gaps” in the scientific knowledge to formulate the aim of their study.

Instead of what was done, Authors should present the aim of the study (e.g. “The aim of the study was…”)

 Corrected:  The aim of the study was   estimate the effect of the consumption of products with excessive amount of critical nutrientes according to the PAHO Nutrient Profile Model (PAHO NPM) on the quality of the diet of children 4 to 12 yrs in Montevideo, Uruguay.

Materials and Methods:

Authors should present necessary details associated with applied methodology, including 24HR (how was it conducted), but also the other stages of the applied procedure

Corrected. The data source for this analysis is the study of the Estimation of Food Intake by multiple pass 24-hour dietary recall [24HR] conducted within the Evaluation of the School Feeding Program and monitoring of the nutritional status of children of public and private elementary schools in Uruguay in 2018 The Research project was conducted by a specially trained Nutritionist Team. During the fieldwork, a professional nutritionist was in the school canteen when the children had their lunch, and observed and quantified the food intake

Authors should describe clearly NOVA classification.

Corrected. The NOVA classification  system groups food according to the nature, extent and purposes of the industrial processed they undergo. This system classifies foods and food products into four groups: group 1 unprocessed and minimally processed foods [G1]; group 2 processed culinary ingredients [G2]; group 3 processed [G3]; group 4 ultra-processed foods [G4] (4,5).

Results:

Authors should present characteristics of the studied participants and they should verify their representativeness

Authors should present the raw data, to describe clearly the studied group

Corrected The sample was carried out by the Instituto Nacional de Estadistica del Uruguay, the children included, as well as the day on which the R24 was to be performed, was random.

The sample was composed of 332 children between 4 to 12 years old. 88 pre-school children (47 male, 41 female), 114 second grade children(64 male, 50 female)  and 130 fifth grade children(65 male, 65 female).

The mean age was: in pre-school 4.5 years old, in second grade 7.9 years old, and in fifth grade was 10.8 years old.

In some tables Authors do not present the data which are indicated in the description – e.g. Mean [95% CI], but there is presented only mean (I suppose), without CI-. Corrected

Authors should not reproduce in the text data which are already presented in tables. Corrected.

 Discussion:

Authors should: (1) compare gathered data with the results by other authors, (2) formulate implications of the results of their study and studies by other authors, (3) formulate the future areas which should be studied.  

Authors' comments  This requires reformulating this section, if you consider that the document cannot be submitted as presented we require postponing this postulation.

The limitations should be broadened and deepened.

 Conclusions: 

The conclusions from the study should be presented. Corrected

In conclusion, this study shows that policies to reduce demand and supply of products, which aim at improving the diet of populations, including school-age children, and protect public health, will be better aligned with those recommended by the WHO if they adopt the PAHO-NPM. Diets free of ultra-processed and processed products with excess free sugars, total fats, saturated fats and sodium were the best option for school-age children in Montevideo, increasing their chances of meeting WHO recommendations. Meanwhile, intake of products with excessive critical nutrients according to PAHO, and each additional gram consumed of such products, significantly worsens diets, preventing them from meeting WHO recommendations.

I send in the document the changes to your comments and suggestions. 
We hope that this second version complies with your comments, 
greetings, atte

Reviewer 3 Report

  1. Terminology (in title and beyond) - is the term "critical nutrients" the best way of describing the focus of this study?
  2. Abstract - remove citations from Abstract to help improve clarity. You should probably refer to intake of dietary energy rather than intake of "calories" to follow standard scientific approaches (consider this revision elsewhere in the manuscript too).
  3. Introduction, paragraph 1 - the brackets around references 7 and 8 appear to be wrong, Should this read (7, 8) or similar?
  4. Introduction, paragraph 4 - this paragraph does not seem to explain the focus of the current research. At the moment, it reads as if you are presenting Results from the current study but the citations given do not align with this. Please consider revising to help rationalise the focus of the current work.
  5. Methods - please rationalise why the current evaluation was only carried out in a sub-sample of the available dataset. Is this a number that would be expected to be representative of the population or wider dataset?
  6. 2.1 Methods - further details of the original study design should probably be described as these do not appear to have been previously published in English. By the other citation (3), it appears that authors may have used a multipass recall method.
  7. 2.2.2 - it would be useful to know where the original source of the ad hoc compositional data came from. Was this e.g. from compositional tables from another country?
  8. Table 1 - in the confidence intervals, is a negative percentage of individuals for the lower boundary appropriate (for data in the first row)? It would seem that a value below 0 can't be considered.
  9. Results page 6, paragraph 4 - to align with the WHO guideline on salt intake, intake in children should probably be considered against a cut-off of 2 g/d adjusted by energy intake in relation to an adult. You can certainly present the average value per day but you may need to rethink how you calculate/present appropriate intake for children across an age range (presumably who have different target energy intakes). https://www.who.int/publications/i/item/9789241504836 
  10. Table 2 - consider defining the two comparator columns more clearly here. "With" and "Without" excessive content... do not seem to describe the different broad dietary patterns clearly.
  11. Table 3 caption - should this start with the word "Assessment".
  12. Paragraph 1, page 7 - the opening sentence of this paragraph is written awkwardly. Nutrient intake would appear to be in excess, not inadequate. please revise for clarity. Please check this page and elsewhere in the manuscript for inappropriate use of the term "inadequate".
  13. Table 4 - I'm unsure of the exact utility of the co-efficients for associations presented here or these data. Consider whether it's worth presenting this information, particularly to such a high number of decimal places. The presence of strong associations is somewhat interesting but perhaps could be presented in-text only. Consider revising or removing table.
  14. Remove the section on Patents at the end of the manuscript.

Author Response

  1. Terminology (in title and beyond) - is the term "critical nutrients" the best way of describing the focus of this study?

Authors' comments we believe that critical nutrients in an appropriate term for this title.

2.Abstract - remove citations from Abstract to help improve clarity. You should probably refer to intake of dietary energy rather than intake of "calories" to follow standard scientific approaches (consider this revision elsewhere in the manuscript too). Corrected

3.Introduction, paragraph 1 - the brackets around references 7 and 8 appear to be wrong, Should this read (7, 8) or similar?

Corrected. We had some problems with the citations, which we have already adjusted.

4.Introduction, paragraph 4 - this paragraph does not seem to explain the focus of the current research. At the moment, it reads as if you are presenting Results from the current study but the citations given do not align with this. Please consider revising to help rationalise the focus of the current work.

Corrected

Previous reports of the evaluation of school feeding program shown that  average group intake in children that participated in , exceeds the daily calorie recommendation by 13%, and 54% of them have an excessive caloric intake, with one out of every three calories consumed from UPP. The average intake of free sugars was close to 100 grams per day, and 2000 mg in the case of sodium. Additionally, children in the study have an insufficient intake  of nutrients  recommended to reach a healthy diet  (2,4,5). School-age children showed an average consumption of fruits and vegetables that did not reach 50% of the recommendation, which is reflected in the low intake of dietary fiber (2,4,5).

5.There is not enough information in Uruguay on how the intake of UPP impacts the nutrient profile in school-age children. In addition, there is insufficient information on the consumption of nutrients linked to NCDs from an early age. Corrected

6.The aim of the study was  estimate the effect of the consumption of products with excessive amount of critical nutrientes according to the PAHO Nutrient Profile Model (PAHO NPM) on the quality of the diet of children 4 to 12 yrs in Montevideo, Uruguay.

7.Methods - please rationalise why the current evaluation was only carried out in a sub-sample of the available dataset. Is this a number that would be expected to be representative of the population or wider dataset?

Corrected The sample was carried out by the Instituto Nacional de Estadistica del Uruguay, the children included, as well as the day on which the R24 was to be performed, was random.

The sample was composed of 332 children between 4 to 12 years old. 88 pre-school children (47 male, 41 female), 114 second grade children (64 male, 50 female)  and 130 fifth grade children(65 male, 65 female).

The mean age was: in pre-school 4.5 years old, in second grade 7.9 years old, and in fifth grade was 10.8 years old.

Note: The data obtained from this study are representative of schoolchildren attending public schools with school canteen service in the city of Montevideo. In Uruguay, one third of school-age children live in this city, the capital of the country, and 80% of them attend public schools. 

8.  2.1 Methods - further details of the original study design should probably be described as these do not appear to have been previously published in English. By the other citation (3), it appears that authors may have used a multipass recall method.

Corrected  DOI: https://doi.org/10.14306/renhyd.26.S2.1337

9.  2.2.2 - it would be useful to know where the original source of the ad hoc compositional data came from. Was this e.g. from compositional tables from another country?

Corrected

For all foods, the old Uruguayan Food Composition Table was used as the first source of information. When it lacked data, the Brazilian Table of Food Composition was used and, thirdly, the Spanish Table of Food Composition.

For preparations commonly consumed by children, the School of Nutrition (EN) of the University of the Republic (UdelaR) analyzed the chemical composition of standardized preparations per 100 grams and added the information from the recipes of the PAE.

In the case of industrialized foods and products, the information provided by the companies on the web or the nutritional composition on the packaging was used.

10. Table 1 - in the confidence intervals, is a negative percentage of individuals for the lower boundary appropriate (for data in the first row)? It would seem that a value below 0 can't be considered.

11.Results page 6, paragraph 4 - to align with the WHO guideline on salt intake, intake in children should probably be considered against a cut-off of 2 g/d adjusted by energy intake in relation to an adult. You can certainly present the average value per day but you may need to rethink how you calculate/present appropriate intake for children across an age range (presumably who have different target energy intakes). https://www.who.int/publications/i/item/9789241504836 

Content of NCD-related critical nutrients: Free sugars, total fat, saturated fats [% of the total energy intake]. Sodium content: total sodium [mg] less the recommended value per age group [2000mg [for adults and adolescents]; 1640mg [for children aged [5-10[years old]; and 1122mg [for children aged <5 years old]]. Guideline: sodium intake for adults and children. World Health Organization; 2012; Human energy requirements. FAO/WHO/UNU; 2004

12. Table 2 - consider defining the two comparator columns more clearly here. "With" and "Without" excessive content... do not seem to describe the different broad dietary patterns clearly.

Corrected Table 2. Mean intakes of NCD-related critical nutrients in the overall population and in fractions of the population consuming products with and without excessive content in NCD-related critical nutrients according to the PAHO NPM

13. Table 3 caption - should this start with the word "Assessment". Corrected

14. Paragraph 1, page 7 - the opening sentence of this paragraph is written awkwardly. Nutrient intake would appear to be in excess, not inadequate. please revise for clarity. Please check this page and elsewhere in the manuscript for inappropriate use of the term "inadequate". corrected

15. Table 4 - I'm unsure of the exact utility of the co-efficients for associations presented here or these data. Consider whether it's worth presenting this information, particularly to such a high number of decimal places. The presence of strong associations is somewhat interesting but perhaps could be presented in-text only. Consider revising or removing table.

Authors' note we considered including the values per 100 grams, but it would not change the decimals much.

16. Remove the section on Patents at the end of the manuscript. done

I send in the document the changes to your comments and suggestions. 
We hope that this second version complies with your comments, 
greetings, atte

Round 2

Reviewer 1 Report

Well done on addressing my concerns.

Author Response

Thanks for your comments.

Reviewer 2 Report

The manuscript entitled “Estimation of intake of critical nutrients associated with noncommunicable diseases according to the PAHO/WHO criteria in the diet of school-age children in Montevideo, Uruguay” presents interesting issue but some problems must be corrected.

Major:

  1. Authors should prepare their manuscript according to the instructions for authors (e.g. Abstract)
  2. The manuscript is shabbily prepared e.g. with the references presented as [4,5,5] – reader does not know which reference was meant by the second “5” (6?, 15?)

Abstract:

Conclusions should be deepen

Introduction:

Authors failed to justify the need for their study – they should present what is already known and what are the “gaps” in the scientific knowledge to formulate the aim of their study.

Materials and Methods:

Authors should present necessary details associated with applied methodology, including 24HR (how was it conducted), but also the other stages of the applied procedure

Authors should describe clearly NOVA classification

Results:

Authors should present characteristics of the studied participants and they should verify their representativeness in comparison with the general characteristics of this age group

Authors should present the raw data, to describe clearly the studied group

Authors should not reproduce in the text data which are already presented in tables

Discussion:

Authors should: (1) compare gathered data with the results by other authors, (2) formulate implications of the results of their study and studies by other authors, (3) formulate the future areas which should be studied.

The limitations should be broadened and deepened.

Conclusions:

The conclusions from the study should be presented

Author Response

Dear reviewer, we appreciate your comments and suggested improvements. In this version we believe that most of them have been resolved. We apologize for some errors in the references. 
One of your final recommendations refers to the need to include data from other studies with similar characteristics, and that part was not included in this version either. This study is part of a series being conducted in multiple countries, and the joint comparative analysis will be published soon. 
If you think it is essential to include the analysis you suggest in this article, please extend the deadline, since in Uruguay it is the time of leaves and vacations and the team is not available, 
I thank you again for your comments, wishing you a very good beginning of the year.
greetings
Florencia

Authors should prepare their manuscript according to the instructions for authors (e.g. Abstract) Done 

  1. The manuscript is shabbily prepared e.g. with the references presented as [4,5,5] – reader does not know which reference was meant by the second “5” (6?, 15?) Fixet 
  2. Conclusions should be deepen

  3. Authors failed to justify the need for their study – they should present what is already known and what are the “gaps” in the scientific knowledge to formulate the aim of their study.

    Authors should present necessary details associated with applied methodology, including 24HR (how was it conducted), but also the other stages of the applied procedure. DONE 

    Authors should describe clearly NOVA classification done 

    Results:

    Authors should present characteristics of the studied participants and they should verify their representativeness in comparison with the general characteristics of this age group done 

    Authors should present the raw data, to describe clearly the studied group 

    Authors should not reproduce in the text data which are already presented in tables done 

    Discussion:

    Authors should: (1) compare gathered data with the results by other authors, (2) formulate implications of the results of their study and studies by other authors, (3) formulate the future areas which should be studied.

    The limitations should be broadened and deepened. 

    The conclusions from the study should be presented. Done 

Reviewer 3 Report

I would like to thank authors for responding to all reviewer comments.

Author Response

Thanks for your comments.